# The Role of ZNF143 in Breast Cancer Cell Survival Through the NAD(P)H Quinone Dehydrogenase 1–p53–Beclin1 Axis Under Metabolic Stress

**DOI:** 10.3390/cells8040296

**Published:** 2019-03-30

**Authors:** A Rome Paek, Ji Young Mun, Mun Jeong Jo, Hyosun Choi, Yun Jeong Lee, Heesun Cheong, Jae Kyung Myung, Dong Wan Hong, Jongkeun Park, Kyung-Hee Kim, Hye Jin You

**Affiliations:** 1Division of Translational Science, Research Institute, National Cancer Center, 323 Ilsan-ro, Ilsandong-gu, Goyang, Gyeonggi 10408, Korea; arpaek@ncc.re.kr (A.R.P.); cromanyon@naver.com (M.J.J.); 2Department of Structure and Function of Neural Network, Korea Brain Research Institute, Daegu 41068, Korea; jymun@kbri.re.kr; 3BK21 Plus Program, Department of Senior Healthcare, Graduate School, Eulji University, Daejeon 34824, Korea; hyokchoi0123@gmail.com; 4Department of Cancer Biomedical Science, National Cancer Center Graduate School of Cancer Science and Policy, National Cancer Center, 323 Ilsan-ro, Ilsandong-gu, Goyang, Gyeonggi 10408, Korea; 74424@ncc.re.kr (Y.J.L.); heesunch@ncc.re.kr (H.C.); jkmyung@ncc.re.kr (J.K.M.); 5Division of Cancer Biology, Research Institute, National Cancer Center, 323 Ilsan-ro, Ilsandong-gu, Goyang, Gyeonggi 10408, Korea; 6Bioinformatics Analysis Team, Research Institute, National Cancer Center, 323 Ilsan-ro, Ilsandong-gu, Goyang, Gyeonggi 10408, Korea; dwhong@ncc.re.kr (D.W.H.); jkpark@ncc.re.kr (J.P.); 7Proteogenomic Analysis Team, Research Institute, National Cancer Center, 323 Ilsan-ro, Ilsandong-gu, Goyang, Gyeonggi 10408, Korea; kyunghee@ncc.re.kr

**Keywords:** ZNF143, p53, NQO1, autophagy, metabolic stress, survival

## Abstract

Autophagy is a cellular process that disrupts and uses unnecessary or malfunctioning components for cellular homeostasis. Evidence has shown a role for autophagy in tumor cell survival, but the molecular determinants that define sensitivity against autophagic regulation in cancers are not clear. Importantly, we found that breast cancer cells with low expression levels of a zinc-finger protein, ZNF143 (MCF7 sh-ZNF143), showed better survival than control cells (MCF7 sh-Control) under starvation, which was compromised with chloroquine, an autophagy inhibitor. In addition, there were more autophagic vesicles in MCF7 sh-ZNF143 cells than in MCF7 sh-Control cells, and proteins related with the autophagic process, such as Beclin1, p62, and ATGs, were altered in cells with less ZNF143. ZNF143 knockdown affected the stability of p53, which showed a dependence on MG132, a proteasome inhibitor. Data from proteome profiling in breast cancer cells with less ZNF143 suggest a role of NAD(P)H quinone dehydrogenase 1(NQO1) for p53 stability. Taken together, we showed that a subset of breast cancer cells with low expression of ZNF143 might exhibit better survival via an autophagic process by regulating the p53–Beclin1 axis, corroborating the necessity of blocking autophagy for the best therapy.

## 1. Introduction

As the most common cancer among women in the world, breast cancer is heterogeneous. Despite accumulated molecular characteristics related to subtypes [1,2,3,4], more effort is required to understand breast cancer for better therapeutics. Two common invasive breast cancers, invasive ductal carcinoma (IDC) and invasive lobular carcinoma (ILC), account for 90% of diagnosed invasive breast cancer patients [5]. The comparison between IDC and ILC has been studied for the possible development of effective treatments. Patients with hormone-responsive (HR)ILC were less likely to receive chemotherapy than those with HR-positive IDC, suggesting distinct therapeutic strategies for IDC and ILC [6]. A molecular understanding based on genetic profiling of breast cancer demonstrated that ILC shows a discohesive morphology by losing CDH1 expression, while IDC retains intact cell–cell adhesion [5]. We have shown that ZNF143 expression in ductal epithelium of normal breast tissues [7], implying a role for ZNF143 expression during the development of IDC.

Autophagy is a cellular process used for recycling amino acids and energy through lysosomes, which maximizes survival when cells are deprived of nutrients or stressed from exposure to drugs [8,9,10]. Autophagy initiates from phagophores, the initial sequestering compartment, which expands into autophagosomes, amphisomes, and autolysosomes by fusion with lysosomes [10] to degrade the contents for further use. The regulatory mechanism of autophagy in cancer has therefore been studied to find possible targetable steps to be used in treatments. However, the role of autophagy in tumorigenesis is complex. Pharmacological or genetic inhibition of autophagy in dormant breast cancer cells has recently been shown to result in decreased cell survival and metastatic burden in mice and human three-dimensional in vitro and in vivo preclinical models of dormancy [11], implying that targeting autophagy stimulation might be beneficial for breast cancer recurrence or metastasis.

Zinc-finger protein 143 (ZNF143), as a human homolog of the Xenopus selenocysteine tRNA gene transcription activating factor [12], has been shown to be essential for normal development in zebrafish via experiments with morpholino antisense oligonucleotides targeting ZNF143 [13]. ZNF143 has also been implied in regulating BUB1B [14], TFAM [15], and GPX1 [16]. Genome-wide analyses have also shown that ZNF143 is a chromatin-looping factor [17,18,19,20], suggesting a role besides being a transcription factor. ZNF143 has been found in numerous cancers, such as lung adenocarcinoma [21], leukemia [20], colon cancer cells [22], prostate cancer [23], gastric cancer [24], and breast cancers [7], implying a role in tumor development. 

In breast tissues, ZNF143 is localized in the ductal epithelium [7]. Interestingly, expression of ZNF143 decreases in tumor tissues during cancer progression, suggesting a correlation between ZNF143 loss and tumor malignancy, especially in ductal epithelium-originated tumors, such as IDC. ZNF143 knockdown has also been shown to increase cell motility in colon cancer cells and breast cancer cells [7,22]; however, it remains unclear whether ZNF143 expression affects tumor malignancy by regulating cell biology more than motility, such as cell viability.

In the present study, we therefore characterized the relationships between ZNF143 expression and cancer cell survival in invasive ductal carcinoma cells under normal and metabolic stresses, such as fetal bovine serum (FBS) or glucose deprivation. Furthermore, we identified the regulatory mechanism involved in how ZNF143 expression affects cell survival.

## 2. Materials and Methods

### 2.1. Cell Culture and Transfection 

Human breast cancer MCF7 and T47D cells was purchased from the American Type Culture Collection (ATCC, Rockville, MD, USA). All cells were authenticated by short-tandem repeat polymerase chain reaction (PCR) method in 2017 (MCF7 and T47D) through NCC Omics Core facility (Perkin Elmer, Waltham, MA, USA). Cells were cultured in RPMI1640 (Gibco, Waltham, MA, USA) containing 10% (*v/v*) fetal bovine serum (FBS) (Gibco, Waltham, MA, USA) and 1% (*v/v*) antimycotic antibiotic (Gibco, Waltham, MA, USA). MCF7 sh-Control, sh-ZNF143 cell lines and T47D sh-Control and sh-ZNF143 cell lines [7] were maintained 1 μg/mL puromycin dihydrochloride (Sigma-Aldrich, St. Louis, MO, USA). 

For transient transfection of ZNF143 (pFLAG-CMV-ZNF143), growing cells were transfected with pFLAG-CMV or pFLAG-CMV-ZNF143FL plasmid using lipofectamine2000 according to the manufacturer’s instructions. After 24 or 48 h, cells were harvested for RT-PCR or immunoblotting.

### 2.2. Live and Dead Cell Staining by Flow Cytometry 

Cells (3 × 10^5^) were grown on 6-well plates for 24 h and exposed to different nutrient deprivations (glucose-free (G−), FBS-free (F−), glucose-free, and FBS-free (G−/F−), or growth media (G+/F+)) for another 24 h. Adherent and nonadherent (dying) cells were harvested using trypsin followed by centrifugation and staining with 0.5 µg/mL propidium iodide (PI; Sigma-Aldrich, St, Louis, MO, USA) in phosphate-buffered saline (PBS). The cells were incubated on ice for 5 min and analyzed by flow cytometry (FACSVerse; BD Biosciences, San Jose, CA, USA). The number of PI positive cells was determined in every 10,000 cells.

### 2.3. Live Imaging for Cell Growth and Survival in Nutrient Deprivation by IncuCyte ZOOM^®^

Cells (3 × 10^3^) were grown on 96-well plates for 24 h and exposed to different nutrient deprivations followed by using the IncuCyte ZOOM^®^ system (Essen Bioscience, Ann Abor, MI, USA) taking images of the same location using a 10× magnification lens, every 2 h for 72 h. Some cells were treated with 10 μM chloroquine or 100 nM Wortmannin, and the relative confluence of each cell at 0 h was analyzed and compared.

### 2.4. Transmission Electron Microscopy (TEM)

Cells were fixed in 0.1 M sodium phosphate buffer (pH 7.4) containing 2.5% (*v/v*) glutaraldehyde at 4 °C. Each sample was washed with the same buffer and fixed for 2 h in 2% (*w/v*) OSO_4_. The cells were subsequently washed and dehydrated using graded alcohol (50% (*v/v*), 70%, 80%, 90%, 95%, and 100%). The cells were then embedded in Spurr media (Sigma-Aldrich, St. Louis, MO, USA) and 60-nm thickness sections were prepared on an ultra-microtome (UC7; Leica Microsystems, Wetzlar, Germany). The sectioned cells were mounted on copper grids and double-stained with uranyl acetate and lead citrate, then observed using a Technai G2 TEM (Thermo Fisher, Waltham, MA, USA).

### 2.5. Autophagosome Measurements 

To measure autophagic vacuoles in live cells, the cells were grown in coverslips and exposed to different nutrient deprivations for another 24 h, then stained with a CYTO-ID^®^ Autophagy Detection Kit 2.0 (Enzo Life Sciences, Farmingdale, NY, USA) [25]. Stained cells were visualized by wide-field confocal microscopy (LSM780; Carl Zeiss, Oberkochen, Germany) at 60× magnification, and the number of autophagosomes (foci) was counted using ImageJ software (National Institutes of Health, Bethesda, MD, USA). 

### 2.6. Immunoblotting 

Cells were washed with PBS and lysed with 2× sample buffer (12.5 mM Tris-HCl (pH 6.8), 2% glycerol, 0.4% sodium dodecyl sulfate (SDS), 1% β-mercaptoethanol, and 0.01% (*w/v*) Bromophenol Blue). Lysates were resolved by SDS-polyacrylamide gel electrophoresis (SDS-PAGE), transferred to Amersham Hybond™ sequencing 0.2 μm PVDF membranes (GE Healthcare, Little Chalfont, UK), and blotted with antibodies against ZNF143 (sc-100983; Santa Cruz Biotechnology, Santa Cruz, CA, USA), Beclin1 (#3738; Cell Signaling Technology, Danvers, MA, USA), ATG5 (#2630; Cell Signaling Technology), ATG12-ATG5 (#2630; Cell Signaling Technology), free ATG12 (#2010; Cell Signaling Technology), LC3B (#2775; Cell Signaling Technology), p62 (610832; BD Biosciences), p53 (sc-126; Santa Cruz Biotechnology), p14ARF (sc-8613; Santa Cruz Biotechnology), p-AKTSer473 (#4060; Cell Signaling Technology), p-AKTThr308 (#2965; Cell Signaling Technology), AKT (#9272; Cell Signaling Technology), and β-actin (sc-69879; Santa Cruz Biotechnology). Immunoreactivity was detected using the Miracle-Star™ western blot detection system (iNtRON Biotechnology, Jungwon, Republic of Korea).

### 2.7. Reverse Transcription Polymerase Chain Reaction (RT-PCR) 

Total RNA was isolated using the RNeasy Mini Kit (Qiagen, Hilden, Germany). Total RNA (5 μg) was reverse transcribed using oligo-dT primers and the SuperScript™ III Reverse Transcriptase (Invitrogen, Carlsbad, CA, USA) according to the manufacturer’s protocol, and PCR was performed using the gene-specific primers (Appendix A).

### 2.8. Proteome Profiling by MASS Spectroscopy and Pathway Analysis

The protein samples were reduced by TCEP and alkylated by iodoacetamide (IAA), precipitated using cold acetone. The precipitated samples were digested with mass spec grade trypsin for 12 h at 37 °C. For TMT6-plex labeling experiments, 100 μg peptide from each cell population was labeled with one of six TMT6-reagents according to manufacturer’s protocol. Then, peptides were fractionated using High pH reversed-phase peptide fractionation kit (Thermo Fisher Scientific, Rockford, IL, USA) for LC-MS/MS analysis. Peptides were analyzed by an Q ExactiveTM hybrid quadrupole-orbitrap mass spectrometer (Thermo Fisher Scientific) coupled with an Ultimate 3000 RSLCnano system (Thermo Fisher Scientific). The peptides were loaded onto a trap column (100 μm × 2 cm) packed with Acclaim PepMap100 C18 resin, separated by the analytical column (EASY-Spray column, 75 μm × 50 cm, Thermo Fisher Scientific), were sprayed into nano-ESI source. Full MS scans were acquired over the range *m*/*z* 350–1400 with mass resolution of 140,000 (at *m*/*z* 200). The AGC target value was 3.00 × 10^6^. The ten most intense peaks with charge state ≥ 2 were fragmented in the higher-energy collisional dissociation (HCD) collision cell with normalized collision energy of 32, and tandem mass spectra were acquired in the Orbitrap mass analyzer with a mass resolution of 35,000 at *m*/*z* 200. Database searching of all raw data files was performed in Proteome Discoverer 2.2 software (Thermo Fisher Scientific). SEQUEST-HT was used for database searching against Swissprot-Homo sapiens database. Database searching against the corresponding reversed database was also performed to evaluate the false discovery rate (FDR) of peptide identification. The database searching parameters included precursor ion mass tolerance 10 ppm, fragment ion mass tolerance 0.08 Da, fixed modification for carbamidomethyl cysteine and variable modifications for methionine oxidation. We obtained an FDR of less than 1% on the peptide level and filtered with the high peptide confidence.

Almost 5000 proteins were profiled and compared. Among them, 177 proteins were selected based on altered ZNF143 expression (more than 1.2-fold or less than 0.83-fold in sh-ZNF143 cells compared to that in sh-Control controls, with a significant *p*-value), and were used for pathway analysis with DAVID functional annotation bioinformatics microarray analysis and Ingenuity Pathway Analysis to identify ZNF143 knockdown-related pathways in breast cancer cells

### 2.9. TCGA Provisional Analyses 

TCGA genomic data mining using cBioPortal (http://www.cbioportal.org/) was performed as described previously [7,26,27,28]. 

### 2.10. Measurement of Intracellular Reactive Oxygen Species (ROS) 

Growing cells were washed with warm PBS, trypsinized, and immediately analyzed for green fluorescence by flow cytometry, as described previously [29]. Before harvest, 2′,7′-dichlorodihydrofluorescein diacetate (H_2_DCFDA, 10 μM) was added for 10 min and the analysis was carried out in a FACSCalibur (Becton–Dickinson, Mountain View, CA, USA) by the NCC FACS operator. The cells were sorted at a rate of approximately 500 cells/s, using saline as the sheath fluid and a 488-nm argon laser beam for excitation. A two-parameter dot-plot of the side light scatter (SSC) and forward light scatter (FSC) of the population was generated, and the DCF fluorescence of 10,000 gated cells was measured using log amplification. The arithmetic geometric mean fluorescence channel (Geo MFC) was derived using the CellQuest^TM^ Pro software. 

### 2.11. Statistical Analysis 

All data are expressed as percentages of the control and shown as means ± S.E. Statistical comparisons between groups were made using Student’s *t*-tests. Values of *p* < 0.05 were considered significant.

## 3. Results

### 3.1. ZNF143 Knockdown Protects Cancer Cells from Death During Nutrient Deprivation in MCF7 Cells 

As a tumor grows, cells within the tumor mass are exposed to various cellular stresses, such as hypoxia, acidosis, and metabolic stress [30]. Here, we first investigated if MCF7 breast cancer cells showed a difference in cell survival when cells were stressed, depending on ZNF143 expression. Growing cells were exposed to nutrient deprivation for 24 h and viable cells were quantified by fluorescence-activated cell sorting (FACS) after propidium iodide (PI) staining (Figure 1A,B). Among 10,000 events, the number of PI positive, dying cells was much lower in MCF7 sh-ZNF143 cells than in MCF7 sh-Control cells when the cells were exposed to glucose-free or FBS-free media for 48 h. To confirm the ZNF143 knockdown effect on cell survival under metabolic stress, cells were grown on 96-well plates for 24 h, then were exposed to nutrient-deprived media, and were monitored using the IncuCyte ZOOM^®^ System. Cell confluency was monitored by capturing images every 2 h for 3 days, then the data were automatically quantitated. MCF7 sh-Control and MCF7 sh-ZNF143 cells showed similar growth up to 3 days in growing media as we previously described [22], while starved cells without FBS or glucose, or both deprived media, showed less growth or survival of sh-Control cells than that of sh-ZNF143 cells (Figure 1C). The ZNF143 knockdown effect on cell survival reached a maximum in FBS-free and glucose-free media, which was reversed by chloroquine, an autophagic flux inhibitor (Figure 1D), but not by Wortmannin, an inhibitor for phosphoinositide 3-kinase (PI3-kinase) (Figure 1E). Because chloroquine inhibits autophagy by increasing lysosome pH [10,31], the ZNF143 knockdown effect on cell survival might result from the autophagic process, downstream of PI3-kinase. 

### 3.2. ZNF143 Knockdown in MCF7 Cells Enhances Autophagic Vacuoles

To further support the autophagic process during cell survival under metabolic stress, MCF7 sh-Control and MCF7 sh-ZNF143 cells were analyzed by TEM (Figure 2A). Numerous autophagic vacuoles were observed in MCF7 sh-ZNF143 cells versus sh-Control cells in growing condition. More active organelles and active autophagic flux were shown in MCF7 sh-ZNF143 cells than in sh-Control cells when cells were starved in FBS-free and glucose-free media implying the effect of ZNF143 knockdown for cell survival. To examine whether ZNF143 knockdown affected the autophagic process, cells were stained with CYTO-ID^®^ (Enzo Life Sciences) before fixation, which measured autophagic vacuoles and monitored autophagic flux [25]. Growing cells were exposed to nutrient deprivation for 24 h in the presence or absence of chloroquine and were then stained with CYTO-ID^®^ and fixed for imaging by confocal microscopy, followed by quantitation of foci (Figure 2B,C). Notably, more foci were observed in sh-ZNF143 cells than in sh-Control cells, similar to the TEM data, implying a relationship between ZNF143 and the autophagic process in breast cancer cells. 

### 3.3. ZNF143 Knockdown Increases the Autophagy-Related Gene, Beclin1, in MCF7 Breast Cancer Cells 

To understand the molecular mechanism involved in how ZNF143 knockdown might be related to autophagy, we examined and compared the expression levels of a few autophagy-related genes, such as beclin1, autophagy-related 5 (ATG5), autophagy-related 12 (ATG12), and microtubule-associated protein light chain 3 (MAP1LC3B, LC3B) [10]. Beclin1, a homolog of ATG6, is an important protein in regulating autophagy, and cell death by binding PI3-kinase or Bcl-2 [32,33]. As expected, MCF7 sh-ZNF143 cells expressed more Beclin1, ATG12, and LC3 than MCF7 sh-Control cells, which was further investigated in cells exposed to nutrient-deprivation as shown in Figure 3. 

### 3.4. ZNF143 Knockdown Reduces Levels of p53 Protein through a Distinct Proteasome-Dependent Pathway in MCF7 Breast Cancer Cells

The Beclin1 protein has been shown to be destabilized by ubiquitination in a p53-dependent manner [34]. We therefore investigated whether Beclin1 might be regulated by p53 in ZNF143 knockdown cells, by comparing the protein levels of p53 in MCF7 sh-Control and sh-ZNF143 cells (Figure 4A). Notably, p53 expression was reduced in MCF7 sh-ZNF143 cells, regardless of FBS deprivation. The levels of p53 mRNA were not altered by ZNF143 knockdown (Figure 4B), while ZNF143 recovery in MCF7 sh-ZNF143 cells reversed the levels of p53 protein (Figure 4C), suggesting a role of ZNF143 on the p53 protein stability at the protein level. The p53 was shown to be degraded through an MG132-dependent pathway involving proteasomes, not the chloroquine-dependent pathway involving lysosomes (Figure 4D). However, we did not observe any ubiquitination of p53 in MCF7 sh-ZNF143 cells in the presence of MG132 (data not shown), implying ubiquitin-independent degradation of p53 during the knockdown of ZNF143.

### 3.5. The NAD(P)H-Quinone Oxidoreductase 1 (NQO1)-p53-Linked Cascade Contributes to Autophagy When ZNF143 Expression Decreases 

To explain how p53 is degraded in a proteasome-dependent manner in MCF7 sh-ZNF143 cells and how ZNF143 expression is linked to proteasome regulation through p53 in breast cancer cells, we conducted mass spectrometry to profile and identify differentially expressed proteins in MCF7 sh-Control and sh-ZNF143 cells (NCC Omicscore). Almost 5000 proteins were profiled, and 177 proteins were selected based on altered ZNF143 expression (more than 1.2-fold or less than 0.83-fold in sh-ZNF143 cells compared to that in sh-Control controls, with a significant p-value), and were used for pathway analysis in breast cancer cells (Figure 5A and Appendix A). Because ZNF143 has been identified as a transcription activating factor for the selenocysteine tRNA gene [12], and is involved in regulating reactive oxygen species (ROS) in cells by glutathione peroxidase 1 (GPX1) expression [16], pathways for “cellular oxidant detoxification”, “cellular response to hydrogen peroxide”, and “xenobiotic metabolic process and response to nutrient” led us to link some proteins of the 177 altered protein pool to ZNF143 in terms of redox regulation and p53 (Figure 5A). NAD(P)H-quinone oxidoreductase 1(NQO1), an enzyme that catalyzes the two electron-reductions of various quinones with NADH or NADPH as an electron donor [35], was listed (Appendix A). NQO1 has been reported to play roles in regulating p53 stability in a ubiquitination-independent mechanism [36,37]. Nuclear factor (erythroid-derived 2)-like 2(NRF2) has shown to be involved in NQO1 expression in redox sensitive manner [38,39]. We therefore hypothesized that NQO1 was a ROS-dependent regulator of p53 during the knockdown of ZNF143. We examined protein levels of NQO1 in MCF7 sh-Control and sh-ZNF143 cells (Figure 5B). NQO1 was downregulated by ZNF143 knockdown, and recovered when ZNF143 expression recovered, implying a role for ZNF143 in NQO1 expression (Figure 5C). In addition, ZNF143 knockdown significantly increased cellular ROS levels (Figure 5D), showing an altered equilibrium for ROS during ZNF143 knockdown. Furthermore, we found prostaglandin reductase 1 (PTGR1) and kynureninase (KYNU), targets of NRF2 [40,41], among the listed proteins from profiling (Appendix A) and observed altered expression of PTGR1 and KYNU in MCF7 cells expressing less ZNF143 (Figure 5E), supporting a role of ZNF143 in the oxidative stress–NRF2 pathway. 

### 3.6. ZNF143 Expression Might Be Important for Disease-Free Survival in Breast Cancer

In terms of breast cancer, the dataset released in 2012 (*n* = 1098) confers overall survival and disease-free survival based on most types of gene alterations, including copy number alteration, mutations, mRNAs, and proteins [42,43]. To determine if ZNF143 expression contributes to tumor malignancy, such as recurrence and metastasis in breast cancer patients, we compared disease-free survival between selected patients with altered ZNF143 expression and the remaining patients in the whole data set (Figure 6A,B). Better disease-free survival was observed when patients overexpressed ZNF143 (>2-fold, at the mRNA level or amplified copy number, *n* = 24), while patients with less ZNF143 expression (less than 2-fold, at the mRNA level, *n* = 14) showed significantly poorer disease-free survival, implying that ZNF143 mRNA expression correlated with tumor malignancy in patients. Furthermore, ZNF143 and NQO1 showed significant co-occurrence (Table 1), and ZNF143 showed significant co-occurrence with TP53 using The Cancer Genome Atlas (TCGA) dataset (TCGA 2012 provisional, c-bioportal.org), suggesting a link between ZNF143 expression and TP53.

## 4. Discussion

The major findings of this study were as follows: (1) ZNF143 knockdown protected cancer cells from cell death under metabolic stress in breast cancer; (2) cell survival by ZNF143 knockdown under metabolic stress was dependent on chloroquine, an inhibitor of the autophagic process; (3) ZNF143 knockdown altered proteins related to the autophagic process; (4) the p53–NQO1 axis was related to ZNF143 expression; and (5) according to the TCGA cohort, ZNF143 mRNA expression was related to disease-free survival of breast cancer patients. Taken together, our results suggested that reduced ZNF143 played a role as a regulator for breast cancer malignancy and that autophagy is deeply associated with cancer survival.

The Early Cancer Trial Lists’ Collaborative Group reported 15-year breast cancer recurrence and survival rates, suggesting a steady relapse rate occurring over approximately 15 years [44]. Another study showed a growing burden of metastatic breast cancer in the USA based on Surveillance, Epidemiology, and End Results registries [45]. How breast cancer dormancy is regulated for cancer recurrence or metastasis is an important question. Cells within the tumor mass are exposed to various cellular stresses, such as hypoxia, acidosis, and metabolic stress [30]. Vera-Ramirez and coworkers recently reported that autophagy might be the critical mechanism for the survival of disseminated dormant breast cancer cells [11], supporting a role for autophagy in recurrence and metastasis. In the present study, we showed that ZNF143 expression might be closely related to disease-free survival, supporting a role for ZNF143 during tumorigenesis. In detail, we showed the effects of ZNF143 on tumor malignancy from motility [7,22] to dormancy in breast cancer cells by showing better survival of breast cancer cells under nutrient-deprivation when ZNF143 expression had been downregulated (Figure 1 and Appendix A). 

Importantly, we explained how ZNF143 expression might be responsible for cell survival by showing the effect of ZNF143 knockdown on the Beclin1–p53–NQO1-linked cascade during autophagy. One thing we needed to clarify was the role of TP53. To support the autophagy process in the presence of decreased amounts of p53 proteins, T47D, another invasive ductal cell line with similar hormone receptor expression to MCF7 [46], showed mild, but better survival under nutrient deprivation when ZNF143 was knocked down. However, chloroquine, an inhibitor of autophagic processes, did not abrogate cell survival in a ZNF143 expression-selective manner, supporting a link between ZNF143 and p53 during autophagy. Another zinc-finger protein, Krüppel-like factor 6 (KLF6) has shown a link to p53 for regulating autophagy for liver regeneration [47]. Both zinc-finger proteins, KLF6 and ZNF143, are implied in regulating autophagy through p53 dependence for cellular process for survival or regeneration, while the detailed mechanisms might be different.

There are several ways proteins can be degraded, including the involvement of proteasomes and lysosomes. Specific short-lived proteins have been shown to be degraded through the 20S proteasome, but not the 26S proteasome, which did not require polyubiquitination for processing [37]. NQO1, as a gatekeeper for the 20S proteasome, has been shown to form a ternary complex with the 20S proteasome and p53, in the presence of NADH [37]. When we characterized the effect of MG132, an inhibitor for proteasomal degradation, on p53 stability in MCF7 sh-ZNF143 cells, we determined that p53 ubiquitination might not be the main mechanism for degradation in cells under ZNF143 knockdown conditions. Enhanced autophagy has been shown to improve the survival of p53-deficient cancer cells under stressful conditions, including nutrient depletion [48], supporting a p53 defect in autophagy, as supported by the data shown in Figure 4. To understand the balancing mechanism between autophagy and proteasome for tumor cell survival, and to apply the mechanism for anticancer therapy, additional studies are now being conducted. 

From pathway analyses, we obtained new insights into how ZNF143 affected cancer cell function. As expected, based on our previous reports [7,22,24], pathways related to cell motility were relevant, such as wound healing, extracellular matrix assembly, actin filament organization, leukocyte migration, epithelial cell morphogenesis, and blood vessel endothelial cell migration. Pathways for “response to nutrient”, “cellular response to starvation”, and “negative regulation of apoptotic process” were also significant, supporting the possibility that ZNF143 knockdown-driven alterations of protein expression might be important for cell survival under metabolic stress, such as nutrient deprivation. It is still unclear how ZNF143 knockdown drives autophagy for cell survival, and if ZNF143 and related genes might be factor for drug sensitization, which are being explored.

The data from this study strongly support a role for ZNF143 during breast cancer malignancy. By modulating the autophagic process, ZNF143 might contribute to cell survival and dormancy related to recurrence and metastasis, to which ZNF143 knockdown-driven proteome alterations were closely related. Although further confirmation should be conducted in the future, we currently suggest that breast cancer patients without TP53 mutations and less ZNF143 and NQO1 might be more sensitive when treated with various autophagic regulators as therapeutic treatments.

## Figures and Tables

**Figure 1 cells-08-00296-f001:**
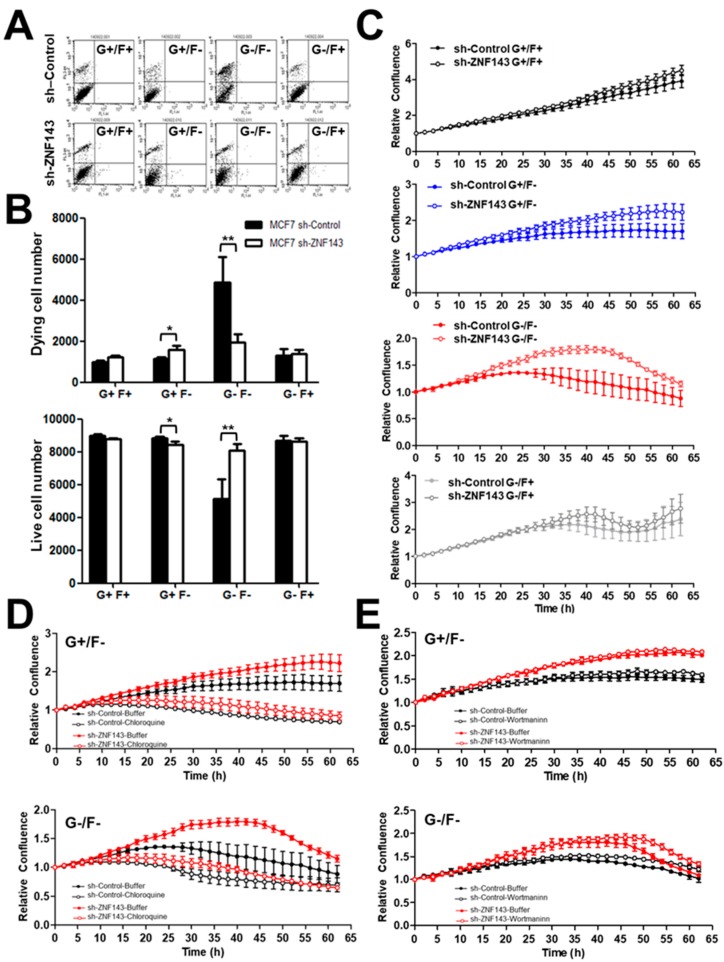
Breast cancer cells with decreased ZNF143 show better survival in glucose- and/or FBS-deprived conditions, which are chloroquine-dependent. (**A**,**B**) MCF7 sh-Control and sh-ZNF143 cells were grown in four different conditions for 24 h, and viable cells were counted by fluorescence-activated cell sorting. G and F denote glucose and FBS, respectively. (**C**–**E**) Cells were plated on 96-well plates and grown for 24 h. The cells were then maintained in four different conditions in terms of FBS and glucose, and cell survival or growth was monitored by capturing images every 2 h over 4 days. Cells were maintained in the presence of 10 μM chloroquine (**D**), 100 nM Wortmannin (**E**), or dimethysulfoxide (vehicle). Relative confluency is shown in the graphs. Data are expressed as means ± S.E. of at least three independent experiments. Statistical significance was assessed using paired Student’s *t*-tests (* *p* < 0.05 and ** *p* < 0.005). Results shown are representative of at least three independent experiments.

**Figure 2 cells-08-00296-f002:**
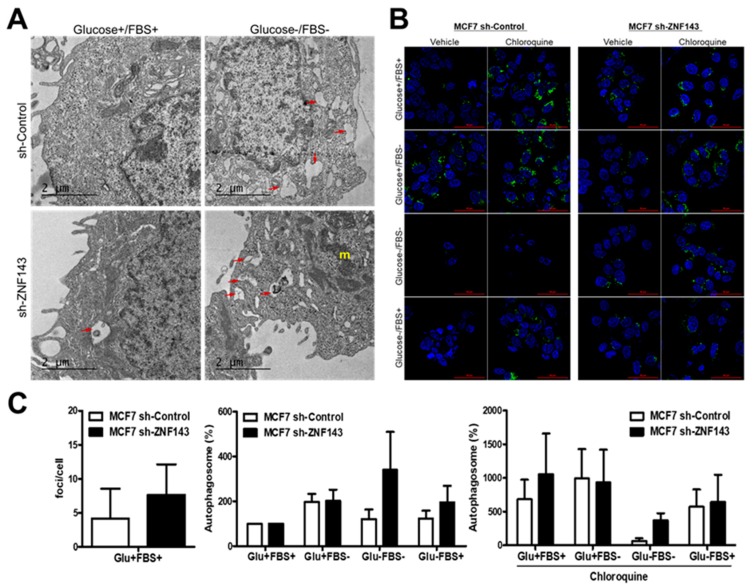
More autophagic vesicles are observed in ZNF143 knockdown cells than in control cells. (**A**) Growing cells were maintained in growing media or fetal bovine serum-free media for 24 h and then fixed for TEM. (**B**,**C**) cells were grown on coverslips for 24 h and then exposed to four different media for an additional 24 h to monitor autophagic processes by autophagosome-selective marker labeling in live cells. Foci were visualized by confocal microscopy, quantified by using ImageJ software, and statistically analyzed by GraphPad software (**C**). Results shown are representative of at least three independent experiments.

**Figure 3 cells-08-00296-f003:**
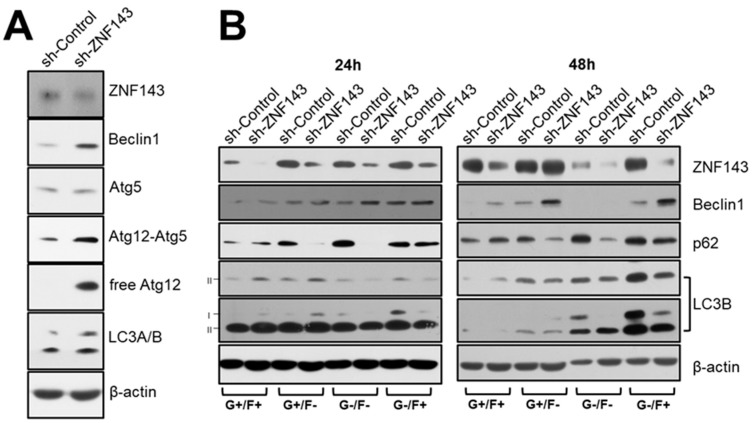
Proteins related to autophagic processes are altered in MCF7 sh-ZNF143 cells. (**A**) Growing cells were harvested for immunoblotting for autophagic-related proteins. (**B**) Growing cells were exposed to four different media (G+/F+, G+/F−, G−/F−, G−/F+; G: glucose, F: FBS, respectively) for additional time periods, as indicated, then harvested for immunoblotting. Results shown are representative of at least three independent experiments.

**Figure 4 cells-08-00296-f004:**
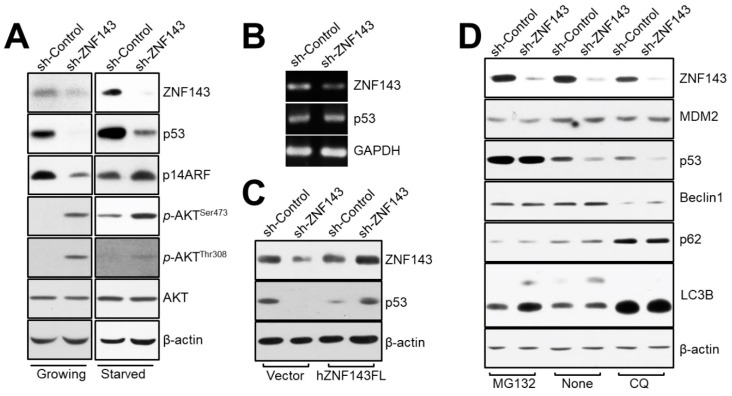
The p53 protein is altered in MCF7 sh-ZNF143 cells in a proteasome-dependent manner. (**A**) Growing or FBS-starved cells were harvested and subjected to immunoblotting. (**B**) Growing cells were harvested for RT-PCR. (**C**) Growing cells were transfected with pFLAG-CMV-hZNF143 or empty vector for 24 h and harvested for immunoblotting. (**D**) Growing cells were harvested for immunoblotting. Cells were treated with MG132 (20 μM, 6 h), chloroquine (50 μM, 18 h), or vehicle. The results shown are representative of at least three independent experiments.

**Figure 5 cells-08-00296-f005:**
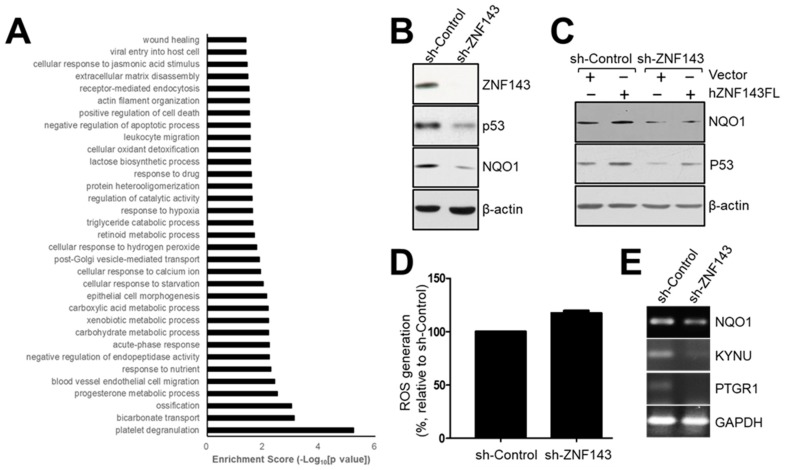
NQO1 is reduced in ZNF143-silenced breast cancer cells, and is important for p53 stability. (**A**) Growing cells were harvested and subjected to liquid chromatography–mass spectrometry to profile altered proteins by ZNF143 knockdown. Approximately 5000 proteins were profiled, and 177 proteins were significantly altered for pathway analysis. (**B**) Growing cells were harvested for immunoblotting to determine the expression of NQO1. (**C**) Growing cells were transfected with a vector (pFLAG-CMV) or a plasmid encoding the full length ZNF143 (hZNF143FL) for 24 h and harvested for immunoblotting. (**D**) Growing cells were harvested for ROS measurements by flow cytometry. Cells were incubated with 10 μM H_2_DCFDA for 10 min before harvesting. (**E**) Growing cells were harvested for RT-PCR to determine the expression of NQO1, PTGR1, and KYNU at the mRNA level. Results shown are representative of at least three independent experiments.

**Figure 6 cells-08-00296-f006:**
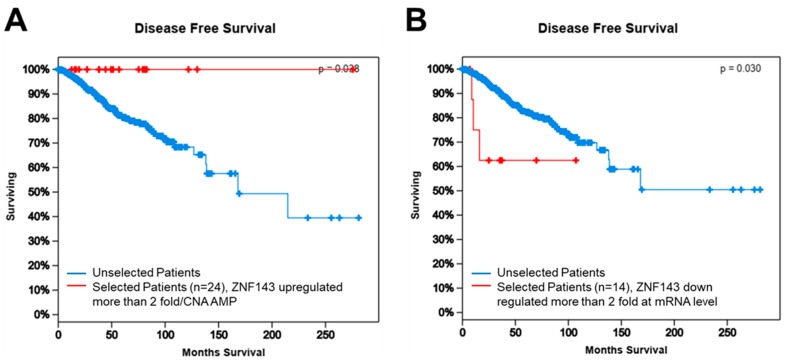
ZNF143 expression is related to disease-free survival of breast cancer patients according to TCGA. (**A**) Higher levels of ZNF143 mRNA expression and copy number alterations were associated with increased disease-free survival in the TCGA cohort. The log-rank value was 0.038. Patients with higher levels of ZNF143 gene expression are shown in red and patients without higher levels of ZNF143 gene expression are shown in blue. (**B**) Lower levels of ZNF143 mRNA expression were associated with decreased disease-free survival in the TCGA cohort. The log-rank value was 0.030. Patients with lower levels of ZNF143 gene expression are shown in red and patients without lower levels of ZNF143 gene expression are shown in blue. In all analyses, patients not applicable for any of the selected attribute(s) were excluded.

**Table 1 cells-08-00296-t001:** Gene Co-occurrent Alteration.

Gene A	Gene B	Neither	A Not B	B Not A	Both	Log Odds Ratio	*p*-Value	Adjusted *p*-Value	Tendency
*NQO1*	*ZNF143*	1019	40	39	7	1.52	0.002	0.007	Co-occurrence *
*ZNF143*	*TP53*	717	22	342	24	0.827	0.005	0.015	Co-occurrence *
*NQO1*	*TP53*	710	29	348	18	0.236	0.267	0.801	Co-occurrence

* adjusted *p*-Value < 0.05, significant.

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
