# Peer review of "The Role of ZNF143 in Breast Cancer Cell Survival Through the NAD(P)H Quinone Dehydrogenase 1–p53–Beclin1 Axis Under Metabolic Stress"

_cells, 2019, doi:10.3390/cells8040296_

Round 1
Reviewer 1 Report
The manuscript of Paek et. al. deals with the role of the zinc finger protein ZNF143 in breast cancer. The group investigated the interaction of autophagy and ZNF143, which has been shown to be associated with many different other cancer types and a potential role in cancer malignancy. The role of ZNF143 was tested in MCF7 breast cancer cells with or without knockout of ZNF143 with the aim to assess the role of ZNF143 within cell survival and to evaluate its potential therapeutic capacity.
The knockout of ZNF143 resulted in an increase of cancer cell survival following starvation and which results from increased autophagy induction. In ZNF143 knockout cancer cells p53 levels were reduced in a proteasome dependent manner. With the use of microarray-derived data the authors identified the protein NQ01 as an important regulating molecule. Analysis of data from patients hints towards a role of ZNF143 in survival and outcome in breast cancer.
The results are reported in a clear and well-organized manner with established and adequate methods and high exactness. The evidence for autophagy induction requires several complex methodological tests, which the authors performed convincingly.
There is a paper, which was published in Scientific Reports in 2017 (Sci Rep. 2017 Aug 14;7(1):8119), that shows the interaction of the zinc finger protein KLF6 with specific autophagy-related molecules in a p53 dependent manner. Perhaps the authors consider the further discussion of this paper in their manuscript.
Author Response
Review Report Form for Reviewer 1
The manuscript of Paek et. al. deals with the role of the zinc finger protein ZNF143 in breast cancer. The group investigated the interaction of autophagy and ZNF143, which has been shown to be associated with many different other cancer types and a potential role in cancer malignancy. The role of ZNF143 was tested in MCF7 breast cancer cells with or without knockout of ZNF143 with the aim to assess the role of ZNF143 within cell survival and to evaluate its potential therapeutic capacity.
The knockout of ZNF143 resulted in an increase of cancer cell survival following starvation and which results from increased autophagy induction. In ZNF143 knockout cancer cells p53 levels were reduced in a proteasome dependent manner. With the use of microarray-derived data the authors identified the protein NQ01 as an important regulating molecule. Analysis of data from patients hints towards a role of ZNF143 in survival and outcome in breast cancer.
The results are reported in a clear and well-organized manner with established and adequate methods and high exactness. The evidence for autophagy induction requires several complex methodological tests, which the authors performed convincingly.
There is a paper, which was published in Scientific Reports in 2017 (Sci Rep. 2017 Aug 14;7(1):8119), that shows the interaction of the zinc finger protein KLF6 with specific autophagy-related molecules in a p53 dependent manner. Perhaps the authors consider the further discussion of this paper in their manuscript.
As pointed out by the reviewer, we added the reference and short discussion in the revised manuscript. Thus, the number of references was increased from 47 to 48 and the reference paper inserted was numbered as 47. The reference 47 in original manuscript became 48 in the revised manuscript. The short discussion related to KLF6 was highlighted yellow in the third paragraph, Discussion Section of revised manuscript. Please check this out.
Reviewer 2 Report
The authors described ZNF143 knockdown protects breast cancer cell survival via NQO1-p53-Beclin 1 cascade under metabolic stress using various biochemical and cell biology methods. Generally it is interesting. But there are several places need to be improve:
1. The authors need to significantly increase the English, some parts, especially in introduction, are difficult for me to understand what the authors try to describe.
2.The authors need to check the format of the chemicals and numbers, for example, in Line 98 3 X 105, in which 5 should be superscript; in Line 115, OSO4 should be OsO4.
3. In Line 220-222, the authors suggested that the effect of ZNF143 knockdown on cell survival might related to the autophagic flux downstream of PI3KC3, why don't you the autophagosome-lysosome legation changes after ZNF143 knockdown this in your results or supplementary?
4.In Line 259, Beclin 1 is a homolog of Atg6 but not Atg6, the way you described would cause confusion. The authors need to check the other protein name description and make sure there is no confusion to readers.
5. All is data described the knockdown effects of ZNF143, I am wondering if you overexposes it, what the effect would be.
Author Response
Review Report Form for Reviewer 2
Comments and Suggestions for Authors
The authors described ZNF143 knockdown protects breast cancer cell survival via NQO1-p53-Beclin 1 cascade under metabolic stress using various biochemical and cell biology methods. Generally, it is interesting. But there are several places need to be improved:
1. The authors need to significantly increase the English, some parts, especially in introduction, are difficult for me to understand what the authors try to describe.
As pointed by the reviewer, we have English Editing service from TEXTCHECK for better description of our finding. Please check the improved expression out in our revised manuscript.
2.The authors need to check the format of the chemicals and numbers, for example, in Line 98 3 X 105, in which 5 should be superscript; in Line 115, OSO4 should be OsO4.
As pointed out by the reviewer, we fixed the format in correct style and highlighted yellow for checkup in the revised manuscript.
3. In Line 220-222, the authors suggested that the effect of ZNF143 knockdown on cell survival might related to the autophagic flux downstream of PI3KC3, why don't you the autophagosome-lysosome legation changes after ZNF143 knockdown this in your results or supplementary?
Based on EM data we have a few idea related to autophagic flux and ZNF143 knockdown, and two students are working on that for our next project. Unfortunately we could not show the data. Please understand our situation.
4.In Line 259, Beclin 1 is a homolog of Atg6 but not Atg6, the way you described would cause confusion. The authors need to check the other protein name description and make sure there is no confusion to readers.
As pointed out by the reviewer, we clarified protein name in the revised manuscript.
5. All is data described the knockdown effects of ZNF143, I am wondering if you overexposes it, what the effect would be.
As suggested by the reviewer, we expressed ZNF143 full ORF in MCF sh-Control cells and sh-ZNF143 cells, which were grown for 48 h and exposed to media without FBS and Glucose both. Cells were observed upto 48 h by live imaging. As shown in figure below, Sh-Control cells showed drastic cell death under severe starvation within 24 h and regardless of ZNF143 expression while sh-ZNF143 cells were shown worse survival when cells were transfected with hZNF143FL (blue box in Figure A bottom panel). The result support the effect of ZNF143 knockdown on cell survival under metabolic stress. Since there are more evidence to support the idea, we did not put the data in the revised manuscript but showed the revision letter for better understanding.
Reviewer 3 Report
The topic is important because to characterize breast cancer cells survival is crucial for therapy. However, the huge amount of data provided here must be better commented in the Discussion to make understandable the final meaning of this study. Please Authors must ameliorate this version as follows:
TITLE: SHORT THE TITLE according to "A role of ZNF143 on breast cancer cell survival through the NAD(P)H quinone dehydrogenase 1-p53-Beclin 1 axis under metabolic stress"
ABSTRACT: At line 39 enlarge the sentence as " ..the p53-Beclin 1 axis, so corroborating the necessity of blocking autophagy for the best therapy."
INTRODUCTION: At line 52 insert in extenso the words corresponding to ZNF143 and also the same must be made at line 69 for BUB1B, TFAM, GPX1...
At line 82 after the word "deprivation" put a point. The start a new sentence as "Furthermore, we identified the regulatory mechanism involved.."
MATERIALS AND METHODS: At line 180 delete in the subtitle 2.10 "by flowcytometry"
RESULTS: From line 197 to line 205 sentences are ina wrong position. I suggest to insert them in the Introduction not here. Please start The Results with the sentence " Here, we first investigated if MCF7 breast cancer cells showed a difference in survival if devoid of ZNF143 when stressed."
At lines 217-222 insert this sentence " The ZNF143 knockdown effect on cell survival reached a maximum in FBS-free and glucose-free media, which was reversed by chloroquine, an autophagic flux inhibitor, (Figure 1D) but not by wortmannin, an inhibitor for phosphoinositide 3-kinase (PI3-kinase) (Figure 1E). Because chloroquine inhibits autophagy by increasing lysosome pH [10,31], the ZNF143 knockdown effect on survival might result from the autophagic process, downstream of PI3-kinase."
At line 235 in subtitle 3.2 put the word "enhances" instead of "results in more". Lines 237-238 rewrite the sentence according to "Numerous autophagic vacuoles were observed in MCF7 sh-ZNF143 cells versus sh-Control cells in growing condition".
Move from line 281 to line 278 the sentence" The p53 was shown to be degraded through an MG132-dependent pathway involving proteasomes." Line 301 please delete too many informations on data analysis like " .. DAVID functional annotation bioinformatics..." and eventually put them in Material and Methods section.
DISCUSSION: At line 354 rewrite the sentence according to " Taken together, our results suggested that reduced ZNF143 played a role as regulator for breast cancer malignancy and that autophagy is deeply associated to cancer survival". Furthermore, enlarge DISCUSSION text according to Results but avoid to insert Figure number here and merely already described results. Finally, readers must better understand why Authors performed this study and its whole meaning.

Author Response
Review Report Form from Reviewer 3
Comments and Suggestions for Authors
The topic is important because to characterize breast cancer cells survival is crucial for therapy. However, the huge amount of data provided here must be better commented in the Discussion to make understandable the final meaning of this study. Please Authors must ameliorate this version as follows:
TITLE: SHORT THE TITLE according to "A role of ZNF143 on breast cancer cell survival through the NAD(P)H quinone dehydrogenase 1-p53-Beclin 1 axis under metabolic stress"
As suggested by the reviewer, we changed the title “A role of ZNF143 on cell survival through the NAD(P)H quinone dehydrogenase 1-p53-Beclin1 axis under metabolic stress in breast cancer cells” into “A role of ZNF143 on breast cancer cell survival through the NAD(P)H quinone dehydrogenase 1-p53-Beclin 1 axis under metabolic stress” and highlighted for review in the revised manuscript.
ABSTRACT: At line 39 enlarge the sentence as " ..the p53-Beclin 1 axis, so corroborating the necessity of blocking autophagy for the best therapy."
As suggested by the reviewer, we added the expression in the revised manuscript. We appreciate the suggestion.
INTRODUCTION: At line 52 insert in extenso the words corresponding to ZNF143 and also the same must be made at line 69 for BUB1B, TFAM, GPX1...
As suggested by the reviewer, we clarified the expression and highlighted for review in the revised manuscript.
At line 82 after the word "deprivation" put a point. The start a new sentence as "Furthermore, we identified the regulatory mechanism involved.."
As suggested by the reviewer, we improved our expression and highlighted for review in the revised manuscript.
MATERIALS AND METHODS: At line 180 delete in the subtitle 2.10 "by flowcytometry"
As suggested by the reviewer, we removed the phrase in the revised manuscript.
RESULTS: From line 197 to line 205 sentences are in a wrong position. I suggest to insert them in the Introduction not here. Please start The Results with the sentence " Here, we first investigated if MCF7 breast cancer cells showed a difference in survival if devoid of ZNF143 when stressed."
Three sentences were removed since the main contents were already described in the Introduction (please see from line 75 to line 80). And the last sentence “as a tumor grows, cells……..metabolic stress [30]” was left to make the paragraph readable, and reasonable for the hypothesis and experiments and highlighted in the revised manuscript.
At lines 217-222 insert this sentence " The ZNF143 knockdown effect on cell survival reached a maximum in FBS-free and glucose-free media, which was reversed by chloroquine, an autophagic flux inhibitor, (Figure 1D) but not by wortmannin, an inhibitor for phosphoinositide 3-kinase (PI3-kinase) (Figure 1E). Because chloroquine inhibits autophagy by increasing lysosome pH [10,31], the ZNF143 knockdown effect on survival might result from the autophagic process, downstream of PI3-kinase."
As suggested by the reviewer, the paragraph was edited and highlighted yellow in the revised manuscript for better description.
At line 235 in subtitle 3.2 put the word "enhances" instead of "results in more". Lines 237-238 rewrite the sentence according to "Numerous autophagic vacuoles were observed in MCF7 sh-ZNF143 cells versus sh-Control cells in growing condition".
As suggested by the reviewer, we improved our expression in the revised manuscript and highlighted in yellow.
Move from line 281 to line 278 the sentence" The p53 was shown to be degraded through an MG132-dependent pathway involving proteasomes." Line 301 please delete too many informations on data analysis like " .. DAVID functional annotation bioinformatics..." and eventually put them in Material and Methods section.
As suggested by the reviewer, sentences from line 298 to 303 were moved to line from176 to line 180 in 2.8. of Materials and Methods Section and modified to describe how we selected the interesting proteins and highlighted yellow in the revised manuscript. And the results now have been focused on the results not methods in the revised manuscript.
DISCUSSION: At line 354 rewrite the sentence according to " Taken together, our results suggested that reduced ZNF143 played a role as regulator for breast cancer malignancy and that autophagy is deeply associated to cancer survival". Furthermore, enlarge DISCUSSION text according to Results but avoid to insert Figure number here and merely already described results. Finally, readers must better understand why Authors performed this study and its whole meaning.
As suggested by the reviewer, Discussion was modified: most figure numbers in discussion were removed except supplementary figure s1 since it appears only once in discussion with Figure 1 in line 364, and highlighted in yellow for better comparison during review process in the revised manuscript. And the sentence was modified as suggested by the reviewer at line 349 and highlighted yellow in the revised manuscript.
Overall we appreciate all comments from the reviewer and we did our best for the review process. Please check the revised manuscript out carefully.
Round 2
Reviewer 2 Report
The authors answered my questions properly and I am satisfied with the answers. But there are some places I suggest the authors to change them:
In Line 204-205, the sentence" Here, we first investigated if MCF7...", you used two ifs in this sentence which still confuses the readers. I think it could be improved.
In Figure 1 D, E, please keep the key to symbols in the same format, either in all four graphs or both in top or bottom graphs.
Table 1 legend, either make the first letter of each word capitalized or only the first letter of the first word capitalized.
Figure 6A, the red line covered the p value, please move the p value to a proper place.
Author Response
Review Report Form for Reviewer 2
Comments and Suggestions for Authors
The authors answered my questions properly and I am satisfied with the answers. But there are some places I suggest the authors to change them:
In Line 204-205, the sentence" Here, we first investigated if MCF7...", you used two ifs in this sentence which still confuses the readers. I think it could be improved.
As pointed by the reviewer, we changed the sentence from “Here, we first investigated if MCF7 breast cancer cells showed a difference in cell survival if devoid of ZNF143 when cells were stressed” to “Here, we first investigated if MCF7 breast cancer cells showed a difference in cell survival when cells were stressed, depending on ZNF143 expression.” and highlighted yellow in the revised manuscript.
In Figure 1 D, E, please keep the key to symbols in the same format, either in all four graphs or both in top or bottom graphs.
As pointed out by the reviewer, we unified the styles through the figures as shown in figure left and replaced in the revised manuscript. Please check this out.
Table 1 legend, either make the first letter of each word capitalized or only the first letter of the first word capitalized.
As shown in the table below, we made the first letter of each word capitalized in Table 1 legend, which was replaced in the revised manuscript. Please check this out.
Figure 6A, the red line covered the p value, please move the p value to a proper place.
Unfortunately, the survival graph was automatically produced from c-bioportal webpage based on the data from selected group and to move p-value was not easy for us. Instead, the figure legends say the log-rank value in detail. “Figure 6. ZNF143 expression is related to disease-free survival of breast cancer patients according to TCGA. (A) Higher levels of ZNF143 mRNA expression and copy number alterations were associated with increased disease-free survival in the TCGA cohort. The log-rank value was 0.038. Patients with higher levels of ZNF143 gene expression are shown in red and patients without higher levels of ZNF143 gene expression are shown in blue. (B) Lower levels of ZNF143 mRNA expression were associated with decreased disease-free survival in the TCGA cohort. The log-rank value was 0.030. Patients with lower levels of ZNF143 gene expression are shown in red and patients without lower levels of ZNF143 gene expression are shown in blue. In all analyses, patients not applicable for any of the selected attribute(s) were excluded.” Thus we think the readers might not be confused.

Reviewer 3 Report
Authors made many efforts to follow referee' criticisms so this version has been greatly ameliorated and now, in my opinion, is suitable for publication.
Author Response
Thank you for your valuable comments, which were very helpful to improve our manuscript.
We all appreciate.